# FocalLens: Instruction Tuning Enables Zero-Shot Conditional Image Representations

## Abstract

Visual feature extraction is fundamental to many vision tasks. Most existing methods extract visual features by encoding an image into a generic feature vector. However, an image naturally contains rich information, and there may be multiple perspectives to describe it. For each application, we might be interested in different aspects of an image and want to prioritize those features over others. For instance, in an image of a dog carrying a toy, if we are primarily interested in the dog, we would expect the extracted features to emphasize the dog over the toy. In this work, we introduce *FocalLens*, a *conditional* visual feature extraction method that produces different representations for the same image based on the context of interest, expressed flexibly through natural language. We leverage vision instruction tuning data and contrastively tune a pretrained vision encoder to take natural language instructions as additional inputs and produce conditional image representations. Extensive experiments validate that conditional image representation from FocalLens better pronounce the visual features of interest compared to generic features produced by standard vision encoders like CLIP. In addition, we show FocalLens further leads to performance improvements on a range of downstream tasks including image-image retrieval, image classification, and image-text retrieval, with an average gain of 5 and 10 points on the challenging SugarCrepe and MMVP-VLM benchmarks, respectively.

## 1 Introduction

Visual feature extraction is a crucial component that underlies many modern vision and multi-modal systems (Ramesh et al., 2021; Li et al., 2022; Liu et al., 2024a; Reid et al., 2024; McKinzie et al., 2024). In recent years, vision foundation models that are pretrained with large-scale datasets (Dosovitskiy, 2020; Chen et al., 2022; Radford et al., 2021; Schuhmann et al., 2022) have become the cornerstone for visual feature extraction, powering downstream applications ranging from classification (Dosovitskiy, 2020), segmentation (Caron et al., 2021), retrieval (Radford et al., 2021), to embodied applications (Driess et al., 2023). Despite the variety of pretraining schemes (Radford et al., 2021; Caron et al., 2021; He et al., 2022; Oquab et al., 2023; El-Nouby et al., 2024), most commonly used vision foundation models, such as CLIP (Radford et al., 2021), are designed to encode the rich information contained in (a patch of) an image into a single feature vector, wherein this *general* feature representation is expected to encapsulate all information that may be leveraged by various potential downstream tasks.

However, by aiming to extract *general-purpose* features that can serve as many downstream tasks as possible, image representations obtained from these task-agnostic vision foundation models may inevitably compromise relevant information that is *specific* to the downstream task of interest. For instance, CLIP models are known to produce image representations that capture the high-level semantics well (Radford et al., 2021; Ramesh et al., 2021), but often struggle with understanding the finer-grained details and intrinsics of the image, such as attribute associations, spatial relationships, camera perspective, and so on (Vaze et al., 2023; Hsieh et al., 2024; Tong et al., 2024b).

In this work, instead of aiming to learn a model that produces a fixed image representation in fulfilling different goals, we consider learning an *adaptive* vision foundation model that encodes an image differently conditioned on the downstream task of interest, allowing the resultant image representations to prioritize information relevant to the specified condition over other available semantics.

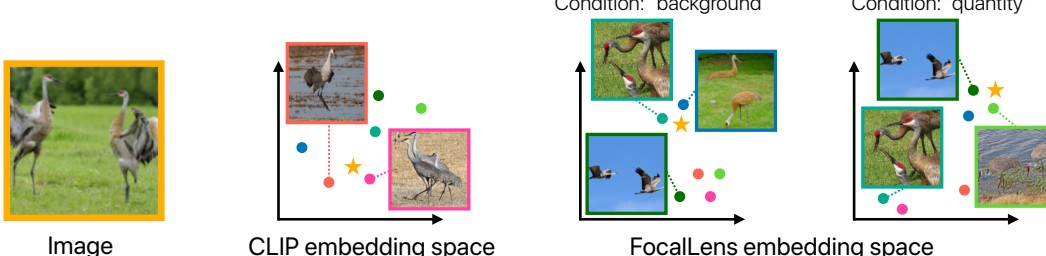

Figure 1: For a given image, the CLIP embedding space is static and structured based on overall semantics. However, FocalLens dynamically rearranges the embedding space based on the specified condition, bringing instances that are more similar under that condition closer together. We show the top-2 nearest neighbors for both CLIP and FocalLens embeddings (once conditioned on "background" and once on "quantity").

Furthermore, as opposed to pre-defining the downstream tasks in a priori (Salehi et al., 2024; Wu et al., 2021), our goal is an *adaptive generalist* model that is able to adapt to broad potential use cases in a *zero-shot* fashion. Specifically, we consider utilizing free-form natural language texts as a rich and flexible interface to condition [1] the model given different downstream purposes, inspired by recent literature (Wei et al., 2021; Su et al., 2022; Liu et al., 2024a). For instance, given a task of retrieving images of similar background scene to a given query image, by specifying through the text condition: "*What is the background in the image?*", we expect to guide the model in focusing more on the background features of the image, as illustrated in Fig. 1.

To develop an adaptive vision foundation model, we start out by first identifying two fundamental characteristics required by the model: (1) the ability to comprehend text conditions, and (2) the ability to produce a corresponding *image representation* with the focus on the given condition. Recent advancement in multi-modal large language models (MLLMs) (Li et al., 2022; 2023; Liu et al., 2024a; McKinzie et al., 2024; Beyer et al., 2024) have shown that these models possess strong capability in understanding both text conditions and images, as demonstrated through their impressive performances on vision question answering tasks. Nonetheless, as these models are inherently trained with the goal of producing next- (text) token predictions, it is unclear whether they could be used to produce *image representations* that generally requires encoding not only the semantics but also other low-level dense features. On the other hand, while existing vision encoders (Radford et al., 2021; Oquab et al., 2023), can produce versatile image representations for various downstream tasks, their representations are fixed and can omit information relevant to specific contexts.

In this work, we propose *FocalLens*, which aligns representations of a pretrained vision-language model (VLM) leveraging instruction tuning data of MLLMs in a contrastive learning manner, to achieve a text-conditional feature extractor to better "focus" on information relevant to the instructions. We take representative pretrained MLLM and vision encoder, LLaVA (Liu et al., 2024a) and CLIP (Radford et al., 2021), as examples, and turn them into conditional feature extractors via FocalLens. We call the resulting models *FocalLens-MLLM* and *FocalLens-CLIP*, respectively, as shown in Fig. 2. In both setups, we use a relatively small vision instruction tuning dataset in the form of (`instruction`, `image`, `output`), as considered in prior MLLM studies (Dai et al., 2023; Liu et al., 2024a). We learn to align the visual representations to respect the specified condition.

Through extensive evaluations on over 60 tasks, we observe that FocalLens models demonstrate a strong ability to condition representations based on the given text instructions, significantly outperforming existing baselines like CLIP. On average, FocalLens achieves up to 9 points higher performance, with even greater improvements on specific tasks, for image-image retrieval tasks. In addition, when used in downstream applications, FocalLens's conditional image representations further lead to clear gains compared to existing baselines. For instance, on image-text retrieval benchmarks, we show an average improvements of 5 and 10 points respectively on SugarCrepe (Hsieh et al., 2024) and MMVP-VLM (Tong et al., 2024b), comparing favorably to other CLIP models that are much larger (up to 2.5×) in size. On image classification, FocalLens also shows superior

---
[1] We use "condition (conditional)" and "adapt (adaptive)" interchangeably in this paper.

performances than CLIP, especially in low-data regime. Finally, further qualitative study showcases various intriguing application scenarios that can be supported by FocalLens.

## 2 RELATED WORK

**Foundation models for vision encoding.** Recently, the computer vision community has witnessed a transformative revolution wherein foundation models pretrained with web-scale datasets (Dosovitskiy, 2020; Jia et al., 2021; Schuhmann et al., 2022; Oquab et al., 2023) are used as the common underlying visual feature extractor to produce versatile image representations that drive various downstream applications (Dosovitskiy, 2020; Radford et al., 2021; Ramesh et al., 2021; Kirillov et al., 2023; Zhou et al., 2022). While there are many pretraining objectives (Oquab et al., 2023; He et al., 2022; El-Nouby et al., 2024; Radford et al., 2021), existing schemes typically train the vision models to produce a single "general" image representation that hopefully captures all relevant information contained in the given image, or utilize information derived from diverse captions to help learning more discriminative image features (Lavoie et al., 2024). Nonetheless, as an image naturally contains rich and dense information, a fixed and general-purpose representation may not sufficiently pronounce information relevant to specific downstream contexts of interest (Kar et al., 2024; Wang et al., 2024; Tong et al., 2024b; Hsieh et al., 2024). These observations motivate our design of a foundation vision encoder that is capable of extracting different representations from a single image conditioned on downstream use cases at test-time, different from universal image embedding approaches that aim to learn a universal model for different domains without explicit conditioning (Google Research, 2023; Ypsilantis et al., 2023).

**Conditional vision representations.** Implicit and task-specific conditioning of visual features have been studied in the literature (Liu et al., 2024a; Dai et al., 2023; Tong et al., 2024a; Eftekhar et al., 2023; Vani et al., 2024; Chameleon Team, 2024). For instance, the hidden representations in MLLMs may be interpreted as a type of conditional image representation, where the visual features are fused with text instructions for producing different output responses (Chameleon Team, 2024). Along this line, InstructBLIP (Dai et al., 2023) further designed a mechanism to extract instruction-aware visual features that are shown to better guide MLLMs to focus on relevant visual features. In the context of embodied AI, conditional (or typically called *selective*) visual representation has also been demonstrated to much improve downstream agent's performances in navigation and displacement tasks (Eftekhar et al., 2023). Nonetheless, conditional visual representations considered in these work are designed and tied in specifically to their model and respective applications. In this work, we are interested in conditional visual representations that may be adopted in broad downstream use cases.

**Vision-language joint representation learning.** There is a rich literature in vision-language (joint) representation learning (Lu et al., 2019; Li et al., 2019; Kim et al., 2021; Radford et al., 2021; Jiang et al., 2024). Our work is related as we aim for a model that can comprehend both images and natural language conditions. Concurrent to our work, E5-V (Jiang et al., 2024) considers MLLM's output space as a universal representation space for both the vision and language inputs. Nonetheless, in addition to the MLLM-based approach, we investigate an alternative promising CLIP-based approach with comprehensive analysis. Relatedly, composed image retrieval (Wu et al., 2021; Saito et al., 2023; Zhang et al., 2024) considers developing models of underlying similar capabilities that generate image embeddings given both image and text. However, different from our goal to use text conditioning to extract downstream-specific *intrinsic* visual features, their goal is to *extrinsically* "compose" semantics from both texts and images, largely towards image-retrieval purposes.

## 3 CONDITIONAL EMBEDDINGS VIA INSTRUCTION CONTRASTIVE TUNING

Our goal is to develop an adaptive vision foundation model that is capable of encoding an image into tailored embeddings conditioned on the downstream task of interest, as specified through natural language texts. Also, we are interested in a conditional representation not tied to specific tasks, but able to generalize to broad instructions.

We consider the visual instruction tuning data (Liu et al., 2024a), which covers diverse tasks, and has demonstrated great generalization of MLLMs in different benchmarks. The visual instruction tuning

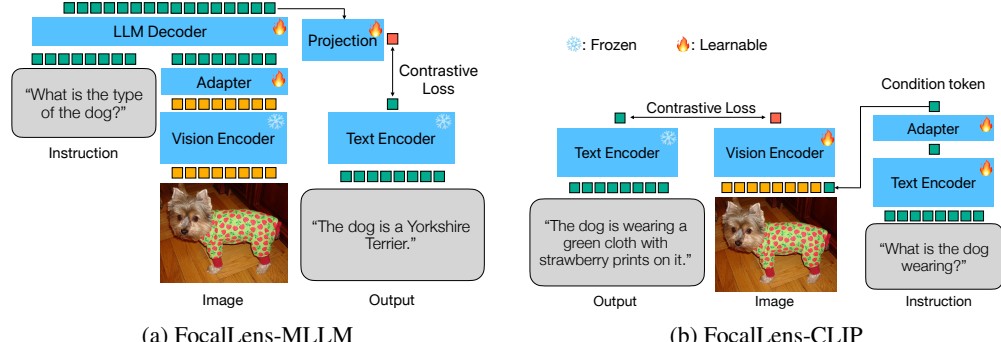

(a) FocalLens-MLLM  (b) FocalLens-CLIP

Figure 2: FocalLens is applied to two vision-language models to extract text-conditioned visual features: **(a)** modifying Llava-like VLMs, which already have text-conditioning capabilities, to produce a global visual feature, and **(b)** modifying ViT (Dosovitskiy, 2020) based CLIP-like VLMs, which already produce a global visual feature, to condition their output feature based on a text condition.

data is in the triplet format of (image, instruction, output). For instance, given an image of "a Yorkshire Terrier wearing a green cloth", the output is "The dog is wearing a green cloth with strawberry prints on it" with the instruction "What is the dog wearing?". Alternatively, when the instruction is "What is the type of the dog", the output is "The dog is a Yorkshire Terrier" correspondingly. MLLMs (Dai et al., 2023; Liu et al., 2024a) leverage the triplet instruction tuning for text generation: given (image, instruction), generating output. Instead, we propose to utilize contrastive learning (Radford et al., 2021) on the triplet instruction tuning data. Specifically, given an image encoder conditioned on the *instruction*, we match the output embedding with a text embedding of *output*. We call the proposed method as *FocalLens*, which leverage instruction tuning data to *contrastively* tune the pretrained image encoder, such that it can better focus on desired information and generalize to diverse downstream tasks. We explore tuning two different representative vision-language models with FocalLens: MLLMs (Section 3.1) and CLIP (Section 3.2).

## 3.1 FOCALLENS WITH MLLMS

MLLMs (Liu et al., 2024a; Dai et al., 2023) generate textual responses regarding an image based on the given input text instructions. Given (instruction, image), the goal is to generate output. However, as the original model objective is text generation rather than producing explicit representation for downstream tasks, the conditional visual information may be dispersed throughout the model, and there is no direct access to them by design.

In FocalLens, instead of training the MLLM to generate output given (image, instruction) as in the original auto-regressive objective, we append a special indicator token <eos_token> to MLLM's input sequence, and consequently train the indicator's output token to align with the CLIP text embedding of the targeted output in a constrative manner. Here, we use an off-the-shelf frozen CLIP text encoder to obtain the target output embedding. With the contrastive objective, we encourage the model to condense information relevant to the image, with the specified instructions, into a single output representation. We show the overall model architecture in Fig. 2a.

## 3.2 FOCALLENS WITH CLIP

Unlike MLLMs, CLIP models by design generate image representations (Radford et al., 2021), where these image embeddings are already widely utilized in a variety of downstream tasks (Ramesh et al., 2021; Liu et al., 2024a). However, CLIP models are inherently limited to producing a fixed representation for each image, regardless of the downstream task of interest. Although strong in capturing high-level semantics, these general visual features are shown to lack various aspects of fine-grained image details that can be critical for downstream tasks (Hsieh et al., 2024; Tong et al., 2024b). To tackle this, we propose to make CLIP's vision encoder task-aware, such that it is able to adapt its representations based on specific requirements, thereby capturing specific aspects of the image essential for different applications.

To incorporate natural language instructions into CLIP's vision encoder, we consider first converting `instruction` into a "condition text embedding", which is then treated as an additional token that is fed into the image encoder alongside the standard image tokens and the CLS token. Afterwards, the model is trained as in standard CLIP using a contrastive loss, aligning the resultant text-conditioned image representations with their corresponding textual outputs. By instruction tuning, we aim to allow the vision encoder to generalize to a broad range of scenarios of interest that can be described via natural language at test-time (Wei et al., 2021; Su et al., 2022). We illustrate the FocalLens-CLIP training setup in Fig. 2b.

## 4 EXPERIMENTS

In this section, we first demonstrate the benefits of conditional image representations (Section 4.1) over the generic representations produced by CLIP, using a toy dataset. We then extensively evaluate FocalLens models' capability in characterizing downstream conditions on a variety of tasks, compared to existing baselines (Section 4.2). By zooming in on FocalLens-CLIP, we demonstrate that its conditional image representations improve performance across a range of downstream tasks, including image-text retrieval, image classification, and image-image retrieval (Section 4.4).

**Setup.** We train FocalLens models with the visual instruction tuning data used in LLaVA (Liu et al., 2024a). The dataset contains around 150k examples, wherein 60k examples are multi-turn conversations and thus can be treated as multiple triplets of (`image`, `instruction`, `output`), where the image remains the same. During training, we expand conversation data within batches to encourage models to output different representations given the same image but different instructions. For FocalLens-MLLM, we follow the training recipe of LLaVA (Liu et al., 2024a) to obtain a base MLLM before further training with the proposed contrastive loss. For FocalLens-CLIP, we initialize the base CLIP model with OpenAI's CLIP-ViT-L-14-336 (Radford et al., 2021), which is also the underlying vision encoder used in LLaVA. We initialize the additional text encoder for instructions to have the same weight as the original text encoder.

For contrastive instruction tuning, given a batch of triplet instruction data $(\boldsymbol{x}_{\text{img}}^{(i)}, \boldsymbol{x}_{\text{ins}}^{(i)}, \boldsymbol{y}^{(i)})$, where $\boldsymbol{y}^{(i)}$ is the expected output for sample $i$, we form the pair-wise similarity matrix $S$, such that

$$S_{i,j} = \phi(\boldsymbol{x}_{\text{img}}^{(i)}, \boldsymbol{x}_{\text{ins}}^{(i)})^T \mathcal{T}(\boldsymbol{y}^{(j)}), \tag{1}$$

where $\phi$ is the encoding process that produce the conditional image embedding from both image $\boldsymbol{x}_{\text{img}}$ and instruction $\boldsymbol{x}_{\text{ins}}$, and $\mathcal{T}$ is the (frozen) text encoder that generates the target embedding from $\boldsymbol{y}$. We apply scaled Softmax to the rows of similarity matrix and compute the contrastive loss following CLIP (Radford et al., 2021). We report further training details in Appendix C. In addition, we report all prompts used for conditioning FocalLens models during evaluation in Appendix D.

**Image-image retrieval as an evaluation protocol.** We consider the common image-image retrieval evaluation to measure the quality of image representations produced from different vision encoders (Google Research, 2023; Caron et al., 2021). Specifically, given a query image, image-image retrieval tasks the model to retrieve other images from a gallery that are "similar" to the query image. We are especially interested in the scenario wherein the very definition of "similar" changes as the downstream tasks vary (Vaze et al., 2023). To facilitate such evaluations, we adopt datasets where we may define various similarities between images based on *test-time* interest determined through a text condition. We introduce these datasets in the following sections. For each dataset, when not otherwise specified, we report mean Average Precision (mAP) as the evaluation metric.

### 4.1 CONDITIONAL REPRESENTATIONS BETTER CHARACTERIZE TASK-SPECIFIC DETAILS

We empirically validate the benefits of having the flexibility to encode an image based on the given condition of interest over using a fixed representation when downstream purpose varies, as considered in most prevailing vision encoding paradigms (Radford et al., 2021; Caron et al., 2021). Here, we restrict ourselves to a toy dataset to demonstrate the idea, and we shall expand our studies in the following sections.

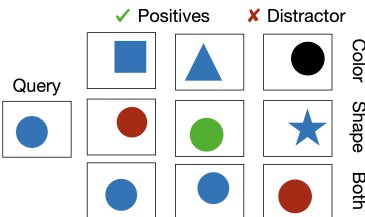

Figure 3: ColorShape examples with a query image, three conditions, and corresponding positives and distractors.

| Model | ColorShape | | | | Cont. Color |
|---|---|---|---|---|---|
| | Color | Shape | Both | Avg. | |
| CLIP (task-agnostic) | 57.10 | 90.24 | 99.36 | 82.23 | 0.158 |
| FocalLens-MLLM | **99.94** | 82.56 | 98.92 | **93.80** | **0.560** |
| FocalLens-CLIP | 87.28 | 93.51 | 99.99 | 93.59 | 0.405 |

Table 1: Image-image retrieval results on ColorShape dataset. Conditional representations from FocalLens better capture the given conditions compared to the task-agnostic representations of CLIP.

**A toy ColorShape dataset.** ColorShape is a synthetic dataset where each image contains a certain colored shape. There are in total 4 different colors and shapes respectively. We generate 500 different images with random position and size of the object for each combination of color and shape. At test-time, we may define the intent for retrieval based on different aspects. Specifically, we may group each image into different categories based on either only its color, only its shape, or both. Fig. 3 shows some examples from the dataset.

The pretrained CLIP model (Radford et al., 2021) serves as the standard encoder baseline where the image representations are fixed even when the test-time condition varies. For the conditional vision encoders, we consider both FocalLens-MLLM and FocalLens-CLIP models discussed in Section 3. We show their retrieval performances on the ColorShape dataset when the test-time condition varies.

**Non-adaptive image representations overlook specific aspects of images.** From Table 1, on the simple ColorShape dataset, CLIP yields almost perfect retrieval performances when we define image categories based on both color and shape. However, in the context where we are specifically interested in categorizing images based only on the color, CLIP's performance drops significantly to 57 mAP point. On the other hand, when we define similarity based only on shape, CLIP achieves relatively better performances at 90 mAP point. Combining the results, while CLIP can produce *general* representation that is strong at grouping objects of certain shape and color together, its overall representation space is biased towards the "shape" of objects, and much less discriminative over the "color" aspect. This also echos the observations made in recent works (Tong et al., 2024b; Hsieh et al., 2024), suggesting that CLIP's representation, while powerful for general tasks, may overlook fine-grained details such as color, highlighting a need for approaches to better adapt and capture the nuanced visual characteristics, depending on the task at hand.

**Conditional image representations better capture information relevant to the downstream task.** In Table 1, as opposed to CLIP model, the conditional image representations produced from both adaptive vision encoders, the MLLM-based and the CLIP-based model, achieve much more balanced (and superior) results than CLIP's representation when the downstream condition varies. When averaged across three different scenarios ("color", "shape", and "both"), both conditional vision encoders improve over 10 mAP point compared to CLIP. The conditional CLIP-based model also always outperforms CLIP, when evaluated separately on the three respective conditions.

In addition to using discrete color labels (e.g., "red", "blue") to define image similarity, we also consider a more sophisticated setup where image similarity is measured based on L2 distance in RGB space. Specifically, in this Continuous Color variant, we assign randomly sampled RGB colors to the objects. During evaluation, our goal is to retrieve images with colors closer to that of the query image. We compute the rank correlation between the similarity measured in the model's image representation space and the ground-truth similarity defined in RGB space. In this setup, both FocalLens models significantly outperform CLIP as show in the last column of Table 1.

### 4.2 FOCALLENS IMPROVES IMAGE REPRESENTATIONS ACROSS BENCHMARKS

Using the ColorShape toy dataset, we validated the benefits of adapting image representations for downstream tasks. We now compare FocalLens to existing vision encoders and relevant baselines across a comprehensive set of evaluation benchmarks.

Table 2: Results on CelebA-Attribute and GeneCIS.

| Model | CelebA-Attribute | | | | | GeneCIS | | |
|---|---|---|---|---|---|---|---|---|
| | Blond Hair | Smiling | Wavy Hair | Lipstick | Avg. 29 tasks | Attribute | Object | Avg. |
| CLIP | 6.20 | 8.68 | 7.54 | 41.45 | 13.59 | 43.10 | 25.81 | 34.46 |
| InstructBLIP | 21.03 | 21.71 | 13.91 | 34.64 | 16.19 | **47.00** | 34.03 | 40.52 |
| MagicLens | 8.24 | 9.98 | 10.76 | 54.12 | 13.42 | 39.00 | 35.50 | 37.25 |
| FocalLens-MLLM | 25.76 | **34.43** | **17.61** | **68.07** | **22.67** | 45.35 | 30.20 | 37.78 |
| FocalLens-CLIP | **32.22** | 22.11 | 16.89 | 62.50 | 21.32 | **43.30** | **43.72** | **43.51** |

Table 3: Results on ImageNet-Subset and fine-grained classification datasets.

| Model | ImageNet-Subset | | | | | Fine-grained classification datasets | | | | |
|---|---|---|---|---|---|---|---|---|---|---|
| | Ball | Cat | Dog | Fish | Avg. 14 tasks | Flower | Car | Aircraft | Food | Avg. |
| CLIP | 64.63 | 53.00 | 16.55 | 61.79 | 51.03 | **83.87** | 45.14 | **25.96** | 58.66 | 53.41 |
| InstructBLIP | 66.44 | 51.22 | 9.60 | 59.16 | 47.67 | 80.26 | 25.97 | 13.47 | 54.32 | 43.51 |
| MagicLens | 68.10 | 50.14 | 17.28 | 58.84 | 46.36 | 74.88 | 23.95 | 17.55 | **65.13** | 45.38 |
| FocalLens-MLLM | **78.99** | 53.24 | 29.25 | 57.40 | 52.34 | 43.92 | 18.59 | 14.73 | 50.93 | 32.04 |
| FocalLens-CLIP | 70.01 | **56.80** | **33.15** | **65.37** | **55.29** | 80.23 | **54.72** | 21.44 | 64.16 | **55.14** |

**Evaluation benchmarks.** We consider a total of 49 different tasks across 4 coarse-grained categories in our evaluation suite as briefly described below. We include dataset details in Appendix A.

- **CelebA-Attribute** (Liu et al., 2015): CelebA is a dataset consisting of celebrity face images. Each face image is associated with various properties spanning from the hair color of the person, the eyebrow shape, to whether the person is wearing eyeglasses, and so on. We vary the downstream condition of interest across different properties for retrieval. For instance, when conditioned on "eyeglasses" with a query image showing a person is (not) wearing eyeglasses, the model is tasked to retrieve other face images with (without) eyeglasses. We manually select a total of 29 different properties that can be objectively labeled, and exclude more subjective properties such as "attractiveness" or "young". We notice that the class within each attribute may be imbalanced, resulting in high mAP even with random guess. We thus report scaled performances w.r.t. random guess by: $\frac{p-r}{1-r}$, where $p$ is the original mAP and $r$ is the random guess mAP.

- **GeneCIS** (Vaze et al., 2023): GeneCIS presents various image retrieval tasks for evaluating conditional image similarity. Given a query image ("a white laptop") and a condition ("color"), the goal is to retrieve the most similar image (another "white laptop") from a gallery that contains implicitly similar distractors with wrong conditions (e.g., "a black laptop"). We report the "Focus attribute" and "Focus object" tasks from GeneCIS. As each query image contains only a single positive in the gallery, we report Recall@3 following prior work (Zhang et al., 2024).

- **ImageNet-Subset** (Deng et al., 2009): In addition to the above benchmarks with specific downstream conditions of interest, we as well evaluate our models on standard ImageNet classes, where the condition corresponds to the image "classes" as defined by ImageNet. Specifically, we create 14 different retrieval sub-tasks based on coarse-grained categories from WordNet (Miller, 1995) hierarchy (e.g., ball, bird, dog, etc.). In each task (e.g., dog), the goal is to retrieve images (from all dog images) with the same type of instance (same breed of dog) as the query image.

- **Fine-grained classification datasets**: Similar to ImageNet, we incorporate 4 finer-grained classification datasets, including Oxford Flowers (Nilsback & Zisserman, 2008), Stanford Cars (Krause et al., 2013), FGVC Aircraft (Maji et al., 2013), and Food-101 (Bossard et al., 2014).

**Baselines.** We consider CLIP (Radford et al., 2021) as the task-agnostic vision encoder model. We also compare to models that are able to generate conditional visual representations, including the Q-former used in InstructBLIP (Li et al., 2023; Dai et al., 2023), and MagicLens (Zhang et al., 2024) that is designed specifically for composed image-retrieval with open-ended instructions. We include details of the baselines in Appendix B.

**FocalLens improves significantly over existing baselines given specific downstream conditions.** From Table 2, both variants of FocalLens provide significant gains over the task-agnostic CLIP

baseline on CelebA-Attribute and GeneCIS, when there are specific conditions to respect. We see an overall gain of 9 points on CelebA-Attribute. Looking more closely at the individual conditions on CelebA-Attribute (complete results reported in Appendix E), we observe that when the condition of interest is "smiling", we see a significant gap of 26 points between CLIP and FocalLens, where the gap is as large as 48 points on certain attributes. Similarly on the GeneCIS benchmark, by specifying the attribute such as color or certain object to focus on, FocalLens improves over CLIP by an average of 9 points.

On CelebA-Attribute and GeneCIS, we also see FocalLens models demonstrate outperforming (or favorable) results when compared to prior task-aware vision encoders (i.e., InstructBLIP and Mag-icLens), that are also given the downstream condition of interest when generating the image representations. Specifically, FocalLens-CLIP achieves the best overall performances, winning over the stronger InstructBLIP baseline by 5 and 3 points respectively on CelebA-Attribute and GeneCIS, validating the effectiveness of our proposed strategy.

**FocalLens maintains or improves over existing baseline on generic conditions.** Here, we compare model performances on ImageNet-Subset and the fine-grained classification datasets, where the downstream goal is generic instance classification. First, CLIP model demonstrates competitive performances on both ImageNet-Subset and fine-grained classification tasks, showing that its embeddings are indeed strong at representing generic features when it comes to standard "type" classification. In contrast, InstructBLIP and MagicLens suffer performance drops on both ImageNet-Subset and fine-grained tasks. On the other hand, we see FocalLens (especially FocalLens-CLIP) maintains comparable performances to CLIP on fine-grained datasets and attains even better performances on ImageNet-Subset. We explain the improvement on ImageNet by that conditioning FocalLens with instructions such as "What is the type of dog?" helps the model to better focus on the specific object of interest but not other potential distractors in the image (e.g., the "toy" besides the dog).

## 4.3 COMPARATIVE ANALYSIS OF FOCALLENS VARIANTS

Both FocalLens-MLLM and FocalLens-CLIP yield promising results in the experiments. One major difference between FocalLens-MLLM and FocalLens-CLIP is their underlying pretrained models' output modality. Specifically, the original MLLM model in FocalLens-MLLM is trained to autoregressively produce textual outputs, while CLIP's vision encoder is trained to produce image embeddings. We are thus interested in understanding whether this difference affects the underlying characteristics of the output representations in FocalLens-MLLM and FocalLens-CLIP.

To test this, we consider downstream conditions that require visual features beyond semantic concepts that are describable by text. In particular, on CelebA, instead of considering conditions such as whether the person is wearing glasses or not, which is answerable in simple words ("yes" or "no"), we consider a *fuzzy* condition where the image similarity is defined by the *identity* of the person. Textual representations that do not carry visual information may fail at achieving good performance on this task, as identity is hardly describable through natural language.

Table 4: Comparison between FocalLens-MLLM and FocalLens-CLIP on fuzzy conditions with CelebA-Identity.

| Model | CelebA-Identity |
|---|---|
| FocalLens-MLLM | 14.48 |
| FocalLens-CLIP | **46.84** |

In Table 4, we observe that FocalLens-MLLM suffers from a clear performance gap compared to FocalLens-CLIP. This suggests that FocalLens-MLLM may rely more on MLLM's original textual output modality, which is limited for tasks requiring rich visual information. Similar observations are also hinted by its relatively low performance on fine-grained classification results in Table 3. In contrast, FocalLens-CLIP, with its underlying model being a vision encoder, is better suited for tasks requiring richer visual detail. Based on this observation, we focus on FocalLens-CLIP for the remainder of the experiments.

## 4.4 FOCALLENS REPRESENTATIONS IMPROVE DOWNSTREAM APPLICATIONS

In addition to evaluations based only on image representations, we show how image representations produced from FocalLens-CLIP can drive improvement on downstream tasks including image-text

retrieval and image classification in a low-data regime where only a small amount of downstream task data is available for training.

**Image-text retrieval.** A prevailing usage of image representations is to enable cross-modality retrieval. Here, we include two image-text prediction benchmarks, where the goal is to predict the correct textual description of a given image. Specifically, we adopt SugarCrepe (Hsieh et al., 2024) and MMVP-VLM (Tong et al., 2024b). SugarCrepe presents challenging hard-negative text distrators along with a positive description for the model to select from, where existing models are shown to struggle with. Similarly, MMVP-VLM particularly collects examples with visual patterns where CLIP vision encoder are shown to fall short.

In Table 6 on SugarCrepe, we compare FocalLens-CLIP to several standard CLIP models of different sizes, and trained with different data sizes. First, compared to the underlying CLIP model used in FocalLens-CLIP (i.e., OpenAI ViT-L-14), FocalLens-CLIP achieves around 4.7 point improvements on average, with consistent improvements across all different sub-tasks with individual gains up to 9 points on Replace-rel and Add-att. Interestingly, the two sub-tasks test the model's capability in understanding fine-grained relationships and attributes in the image, where standard CLIP models struggle the most (Hsieh et al., 2024). This suggests FocalLens-CLIP's image representations are able to better characterize fine-grained visual details. Furthermore, by scaling up the model size from 428M to 623M, the RN50x64 model still underperform our smaller FocalLens-CLIP model (551M for both image and text encoders). On the other hand, FocalLens-CLIP shows competitive performances compared to the $2.5\times$ bigger ViT-g-14 model trained on $5\times$ more data.

From Table 7 on MMVP-VLM, we see FocalLens-CLIP significantly outperforms the baseline ViT-L-14 model consistently across all sub-tasks, by an average of 9.7 points. Furthermore, we note that our FocalLens-CLIP model also compares favorably to the much larger ViT-H-14 ($1.8\times$ larger) and ViT-g-14 ($2\times$ larger) on individual sub-tasks, where FocalLens-CLIP achieves the best overall performance with a lead of 5.2 point.

**Linear probing in low-data regime.** We evaluate the performance of FocalLens-CLIP in a linear probing setup, where only a small amount of downstream task data is available for training. We use the largest dataset in ImageNet-Subset introduced in Section 4.2, focusing on different dog breeds (a total of 118 classes). In the low-data setup (Henaff, 2020; Luo et al., 2017; Vemulapalli et al.), we assume there are $k$ instances available for each class for training and consider $k = 5, 10, 15$. We freeze the backbone and replace the CLIP projection layer with a linear layer to perform 118-way classification. The linear probe is trained for 100 epochs following prior works like (Liu et al., 2024b). We sweep over learning rates from 1e-2 to 1e-4 in steps of 2.5e-3 and report the performance of the best checkpoint. We compare FocalLens-CLIP to OpenAI ViT-L-14 in this setup, as shown in Table 5. In the extreme setting, where only 5 instances per

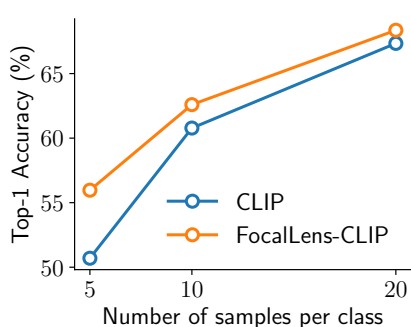

Table 5: Linear probing results comparing CLIP and FocalLens-CLIP.

class is available to train the linear probe, FocalLens-CLIP outperforms CLIP-ViT-L by 5.3%. This result further reinforces our observation that conditional image representations are more efficient in extracting information relevant to downstream tasks.

**Qualitative analysis on conditional image-retrieval.** We qualitatively compare the top-$k$ images retrieved by using FocalLens-CLIP's conditional image embeddings with those retrieved by standard CLIP, specifically when given various downstream conditions. For this qualitative study, we treat all images in the 14 coarse-grained categories considered in ImageNet-Subset as the gallery for retrieval. In Fig. 4, we showcase several intriguing examples across various aspects of conditioning FocalLens-CLIP captures. In the top-left example, we consider a scenario where we are interested in retrieving images of similar background to the query image. Given the query image of "a goose on a grassy field", although the images retrieved by CLIP do all contain goose, all images have the background of water instead of grassy field. Conversely, we see images retrieved by FocalLens-CLIP all have similar grassy background as expected. Similarly, in the top-right, we see FocalLens-CLIP

Table 6: Image-Text Retrieval on SugarCrepe for vision-language compositionality evaluation.

| Model | SugarCrepe | | | | | | | |
|---|---|---|---|---|---|---|---|---|
| | Replace-obj | Replace-att | Replace-rel | Swap-obj | Swap-att | Add-obj | Add-att | Avg. |
| OpenAI ViT-L-14 (2021) | 94.49 | 80.58 | 66.78 | 64.08 | 62.46 | 80.74 | 74.27 | 74.77 |
| OpenAI RN50x64 (2021) | 94.49 | 83.50 | 70.63 | 61.79 | 66.67 | 83.27 | 73.99 | 76.33 |
| LAION ViT-g-14 (2022) | **95.76** | **85.03** | 72.40 | 63.01 | **71.17** | **91.51** | 82.08 | **80.14** |
| FocalLens-CLIP | 95.64 | 84.51 | **75.53** | **65.30** | 66.36 | 86.12 | **83.09** | 79.51 |

Table 7: Image-Text Retrieval on MMVP-VLM.

| Model | MMVP-VLM | | | | | | | | | |
|---|---|---|---|---|---|---|---|---|---|---|
| | Orientation | Presence | State | Quantity | Spatial | Color | Structure | Text | Camera | Avg. |
| OpenAI ViT-L-14 (2021) | 6.7 | 20.0 | 26.7 | 6.7 | 13.3 | 33.3 | **46.7** | 20.0 | 13.3 | 20.7 |
| MetaCLIP ViT-H-14 (2023) | 6.7 | 13.3 | **60.0** | 13.3 | 6.7 | 53.3 | 26.7 | 13.3 | **33.3** | 25.2 |
| EVA01 ViT-g-14 (2023) | 6.7 | 26.7 | 40.0 | 6.7 | 13.3 | 66.7 | 13.3 | 13.3 | 20.0 | 23.0 |
| FocalLens-CLIP | 6.7 | **33.3** | 33.3 | **40.00** | **26.7** | 66.7 | 20.0 | **26.7** | 20.0 | **30.4** |

faithfully reflects the interested condition of quantity, retrieving images with 3 dogs as in the query image, whereas images retrieved by CLIP is largely based on their instance type (same species of dog), and cannot reflect the downstream interest. More examples demonstrate that color or even implicit visual features such as camera angle can also be characterized by FocalLens-CLIP.

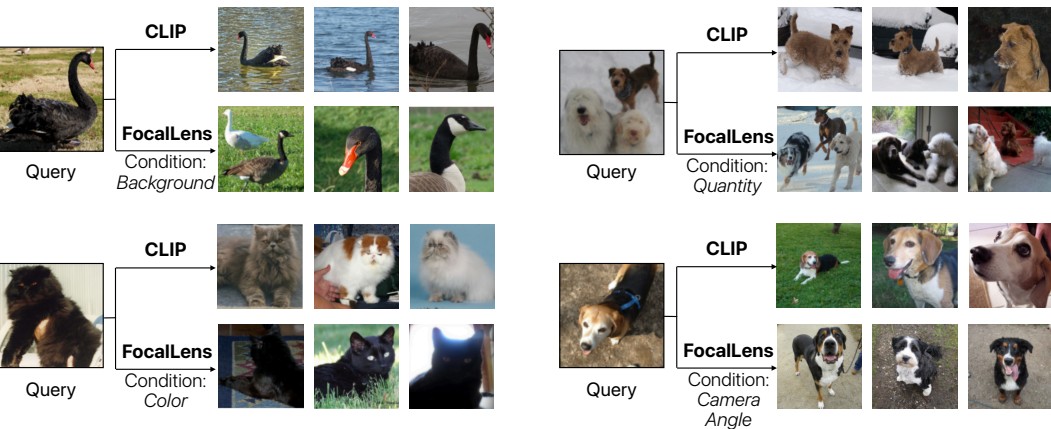

Figure 4: Comparison between CLIP and FocalLens-CLIP on conditional image retrieval.

## 5 CONCLUSION

In this work, we introduced FocalLens, a zero-shot conditional visual embedding model that focuses the representation on specific aspects of the image described in the given text. FocalLens is trained using existing visual instruction tuning datasets to align the conditional image representation with the textual description. Experiments on a comprehensive set of tasks, including image-to-image retrieval, image classification, and image-to-text retrieval, demonstrate that FocalLens matches or exceeds the performance of state-of-the-art models.

**Limitations.** Although experiments demonstrate that FocalLens can be effectively trained using visual instruction tuning datasets, model performance could be enhanced by designing customized datasets for this task, which we leave for future study. Moreover, the relatively small scale of the visual instruction tuning datasets may hinder alignment accuracy for highly specialized concepts that are entirely absent from the dataset.

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

# A DATASETS

**CelebA-Attribute.** There are a total of 40 different binary attributes in CelebA dataset (Liu et al., 2015), from which we select 29 attributes we consider objective, including: "Arched Eyebrows", "Bags Under Eyes", "Bald", "Bangs", "Black Hair", "Blond Hair", "Brown Hair", "Gray Hair", "Blurry", "Bushy Eyebrows", "Double Chin", "Eyeglasses", "Goatee", "Male", "Mouth Slightly Open", "Mustache", "No Beard", "Oval Face", "Pale Skin", "Rosy Cheeks", "Sideburns", "Smiling", "Straight Hair", "Wavy Hair", "Wearing Earrings", "Wearing Hat", "Wearing Lipstick", "Wearing Necklace", "Wearing Necktie".

**ImageNet-Subset.** The ImageNet dataset (Deng et al., 2009) is organized according to the nouns in the WordNet hierarchy (Miller, 1995) and consists of 1000 classes. To evaluate the performance of conditioned representations, we form multiple subsets of ImageNet using the intermediate nodes from the WordNet hierarchy. We list all the ImageNet subsets we created in Table 8.

Table 8: ImageNet-Subset datasets and number of classes per each.

| Node Name | Dog | Bird | Musical Instrument | Snake | Fish | Monkey | Ball | Car | Edible Fruit | Beetle | Cat | Spider | Bag | Piano |
|---|---|---|---|---|---|---|---|---|---|---|---|---|---|---|
| **Num classes** | 118 | 59 | 28 | 17 | 16 | 13 | 10 | 10 | 10 | 8 | 7 | 6 | 5 | 2 |

# B BASELINES

**CLIP.** We consider CLIP as a task-agnostic vision encoder baseline. In all experiments, we use OpenAI's CLIP-ViT-L-patch14-336 released checkpoint (Radford et al., 2021). The model size is 428M including both vision and text encoder. We consider the same model checkpoint in FocalLens-MLLM and FocalLens-CLIP.

**InstructBLIP.** InstructBLIP (Dai et al., 2023) is a MLLM that connects a frozen vision encoder, CLIP (Fang et al., 2023), to a large language model (LLM) decoder to enable multi-modal capabilities. Specifically, it adopts an instruction-aware Q-former architecture (Li et al., 2023) as the connector. The Q-former takes in as input the image embedding extracted from the underlying vision encoder, along with tokenized text instructions. Through cross-attention design, the Q-former outputs multiple instruction-aware image tokens to be fed into the LLM decoder. In our experiments, we average over all image tokens to obtain the image representation used in our evaluations. We use the same instructions as in FocalLens for conditioning InstructBLIP.

**MagicLens.** MagicLens (Zhang et al., 2024) is a model trained specifically for composed image retrieval with a web-scale 36M-sized dataset. The model takes in both a reference image and natural language text to produce image representations that composes the semantics from both the input image and text. In our experiments, we condition MagicLens model using the same text instructions used for FocalLens.

# C EXPERIMENT DETAILS

**Computation resource.** We train FocalLens models on single node machines with 8 A100 GPUs.

**Hyperparameters.** For contrastive training with FocalLens, we report the hyperparameters used in Table 9.

Table 9: Training hyperparameters.

| Model | Batch size | Epoch | Learning rate | Weight decay | Warmup ratio |
|---|---|---|---|---|---|
| FocalLens-MLLM | 384 | 2 | 2e-5 | 0. | 0.03 |
| FocalLens-CLIP | 2048 | 20 | 2e-5 | 0 | 0.03 |

## D   INSTRUCTIONS USED FOR DIFFERENT TASKS

Here, we detail the instructions we use for different tasks for conditioning FocalLens and other instruction-aware baselines.

Table 10: Instructions and templates used for different datasets and conditions.

| Dataset | Condition | Instruction |
|---|---|---|
| ColorShape | Color | What is the color of the object in the image? |
| | Shape | What is the shape of the object in the image? |
| | Both | What is the color and shape of the object in the image? |
| CelebA-Attribute | Noun attributes (e.g., Arched Eyebrows) | Does the person in the image have {attribute}? |
| | Adjective attributes (e.g., Bald) | Is the person in the image {attribute}? |
| CelebA-Identity | - | Gender, age, eye color, hair color, face shape, facial hair of the person. |
| GeneCIS | Focus attribute | Focus on the {attribute}. |
| | Focus object | Is there {object}? |
| ImageNet-Subset | category (e.g., dog) | What type of {category} is in the image? |
| Fine-grained datasets | category (e.g., flower) | What type of {category} is in the image? |
| SugarCrepe | Replace-obj | Focus on the presence of objects in the image. |
| | Replace-att | Focus on the color, patterns and other attributes of the objects in the image. |
| | Replace-rel | What are the relationships between the objects in the image? |
| | Swap-obj | What are the actions, states, colors, patterns and relationships of the objects in the image? |
| | Swap-att | What kind of objects are in the image? |
| | Add-obj | What is not in the image? |
| | Add-att | What is not in the image? |
| MMVP-VLM | Orientation | Describe the orientation, position, or the direction of the object. |
| | Presence | Focus on the presence of objects in the image. |
| | State | Focus on the specific state or the condition of the objects in the image. |
| | Quantity | Focus on the quantity of the objects in the image. |
| | Spatial | Describe the spatial relationship and the positions of the objects in the image. |
| | Color | Focus on the color of the objects in the image. |
| | Structural | Describe the state of the objects in the image. |
| | Text | Focus on the texts on the objects in the image. |
| | Camera | Describe the perspective and view from which the photo is taken. |

## E   FULL EXPERIMENT RESULTS

### E.1   CELEBA-ATTRIBUTE FULL RESULTS

We report full CelebA-Attribute results in Table 11.

Table 11: Full results on CelebA-Attribute.

| Model | Arched Eyebrows | Bags Under Eyes | Bald | Bangs | Black Hair | Blond Hair | Blurry | Brown Hair | Bushy Eyebrows | Double Chin |
|---|---|---|---|---|---|---|---|---|---|---|
| CLIP | 8.13 | 12.00 | 24.52 | 2.86 | 7.96 | 6.20 | 5.52 | -0.58 | 11.98 | 18.35 |
| InstructBLIP | 7.12 | 8.35 | 27.40 | 4.95 | 9.50 | 21.03 | 14.67 | -0.81 | 3.73 | 11.01 |
| MagicLens | 11.32 | 12.10 | 15.14 | 2.44 | 7.48 | 8.24 | 8.95 | -3.22 | 6.75 | 13.88 |
| FocalLens-MLLM | 15.15 | 14.98 | 19.23 | 4.38 | 17.95 | 25.76 | 6.14 | 4.44 | 6.88 | 15.37 |
| FocalLens-CLIP | 13.38 | 13.00 | 26.68 | 8.19 | 10.24 | 32.22 | 11.03 | 5.53 | 9.99 | 15.94 |

| Model | Eyeglasses | Goatee | Gray Hair | Male | Mouth Slightly Open | Mustache | No Beard | Oval Face | Pale Skin | Rosy Cheeks |
|---|---|---|---|---|---|---|---|---|---|---|
| CLIP | 17.84 | 20.16 | 24.19 | 54.55 | 4.72 | 20.92 | 27.64 | 1.63 | 3.22 | -3.15 |
| InstructBLIP | 41.83 | 16.17 | 22.56 | 43.66 | 12.87 | 19.16 | 23.75 | 0.77 | 2.73 | -3.45 |
| MagicLens | 15.52 | 11.28 | 20.13 | 64.56 | 6.04 | 13.50 | 27.52 | 1.83 | 1.98 | 1.95 |
| FocalLens-MLLM | 47.72 | 20.96 | 22.40 | 96.82 | 33.41 | 19.30 | 34.30 | 1.66 | 1.36 | 5.85 |
| FocalLens-CLIP | 24.90 | 29.04 | 23.86 | 95.04 | 10.82 | 26.59 | 41.80 | 0.94 | 4.58 | -0.90 |

| Model | Sideburns | Smiling | Straight Hair | Wavy Hair | Wearing Earrings | Wearing Hat | Wearing Lipstick | Wearing Necklace | Wearing Necktie | |
|---|---|---|---|---|---|---|---|---|---|---|
| CLIP | 18.21 | 8.68 | 3.47 | 7.54 | 7.32 | 17.60 | 41.45 | -0.67 | 21.81 | |
| InstructBLIP | 12.10 | 21.71 | 3.17 | 13.91 | 13.51 | 45.11 | 34.64 | 1.94 | 36.56 | |
| MagicLens | 11.54 | 9.98 | 2.84 | 10.76 | 10.92 | 21.19 | 54.12 | 3.58 | 16.97 | |
| FocalLens-MLLM | 20.02 | 34.43 | 4.50 | 17.61 | 21.54 | 34.32 | 68.07 | 5.05 | 37.86 | |
| FocalLens-CLIP | 32.35 | 22.11 | 2.81 | 16.89 | 12.39 | 33.58 | 62.50 | 3.07 | 29.80 | |

### E.2   IMAGENET-SUBSET FULL RESULTS

We report full ImageNet-Subset results in Table 12.

Table 12: Full results on ImageNet-Subset.

| Model | Bag | Ball | Beetle | Bird | Car | Cat | Dog |
|---|---|---|---|---|---|---|---|
| CLIP | 55.61 | 64.63 | 51.84 | 66.72 | 57.73 | 53.00 | 16.55 |
| InstructBLIP | 60.13 | 66.44 | 51.10 | 45.86 | 60.54 | 51.22 | 9.60 |
| MagicLens | 53.22 | 68.10 | 43.37 | 51.69 | 54.15 | 50.14 | 17.28 |
| FocalLens-MLLM | 63.95 | 78.99 | 41.44 | 54.14 | 54.46 | 53.24 | 29.25 |
| FocalLens-CLIP | 59.44 | 70.01 | 46.88 | 64.62 | 61.84 | 56.80 | 33.15 |

| Model | Fruit | Fish | Monkey | Music Instrument | Piano | Snake | Spider |
|---|---|---|---|---|---|---|---|
| CLIP | 60.95 | 61.79 | 37.79 | 39.18 | 61.97 | 32.03 | 54.61 |
| InstructBLIP | 49.74 | 59.16 | 27.96 | 41.44 | 66.17 | 26.45 | 51.61 |
| MagicLens | 57.40 | 58.84 | 26.82 | 41.18 | 57.40 | 25.74 | 43.76 |
| FocalLens-MLLM | 65.98 | 57.40 | 34.81 | 57.83 | 57.14 | 29.47 | 54.69 |
| FocalLens-CLIP | 69.78 | 65.37 | 38.30 | 61.29 | 60.60 | 32.06 | 53.93 |

