# OpenReview forum: "FocalLens: Instruction Tuning Enables Zero-Shot Conditional Image Representations"
_ICLR.cc/2025/Conference — Submitted to ICLR 2025_

### Official Review · Reviewer_MEKq · 2024-10-29

**Soundness:** 3
**Presentation:** 3
**Contribution:** 2
**Rating:** 5
**Confidence:** 3

**Summary:**

This paper introduces FocalLens which is able to produce different visual representations of the same image based on different contexts of interest. The authors leverage the instruction tuning data, which is usually in the triplet format of (image, instruction, output), to fine-tune MLLMs and CLIPs in a contrastive manner, resulting in FocalLens-MLLM and FocalLens-CLIP, respectively. Evaluations on various benchmarks demonstrate the effectiveness of the propose method.

**Strengths:**

1. Extracting visual features according to specific conditions, e.g., text instructions, is worth studying.
2. Overall, this paper is well-written and easy-to-follow.
3. FocalLens-CLIP appears to be effective.
4. The motivation is clear, and the training pipeline of FocalLens-CLIP is reasonable.

**Weaknesses:**

1. FocalLens-MLLM is somewhat weird. This paper aims to produce context-aware visual features. However, there appears to be a discrepancy in the design, as the visual features produced by FocalLens-MLLM do not seem to be modulated by contextual instructions. Notably, the architecture does not incorporate instructions as inputs to the visual encoder. Consequently, this suggests that the visual features extracted remain invariant across different instructions. Could you explain in more detail how the instruction information modulates the visual features in FocalLens-MLLM? Is there a mechanism that allows the visual encoder to produce different representations based on different instructions?
2. Equipping FocalLens-CLIP with standard MLLM training recipes seems to be an appropriate design. I am curious about the performance. Have you evaluated FocalLens-CLIP's performance on standard multimodal comprehension tasks compared to baseline models? Could you provide results or analysis demonstrating how the context-aware visual features contribute to improved multimodal understanding?

**Questions:**

1. Does the training of FocalLens-MLLM still have the next-token-prediction loss based on cross-entropy?

---

> ### Author Response · Authors · 2024-11-20
> **Response to Reviewer MEKq**
>
> **Question 1:** FocalLens-MLLM is somewhat weird. This paper aims to produce context-aware visual features. However, there appears to be a discrepancy in the design, as the visual features produced by FocalLens-MLLM do not seem to be modulated by contextual instructions. Notably, the architecture does not incorporate instructions as inputs to the visual encoder. Consequently, this suggests that the visual features extracted remain invariant across different instructions. Could you explain in more detail how the instruction information modulates the visual features in FocalLens-MLLM? Is there a mechanism that allows the visual encoder to produce different representations based on different instructions?
> **Response 1:** Thank you for pointing out this confusion. We clarify that in FocalLens-MLLM, the context-aware visual features are produced by the LLM decoder, which is conditioned on both the input image and the input instruction (in Figure 2a). In this case, the entire MLLM is considered a vision encoder, where the fusion between visual features and text instructions happen in the LLM decoder. This is by no means an efficient design, but serves as an exploration and a baseline approach on how we can extract context-aware visual features from existing MLLM built for text generations. We show in our experiments that our proposed FocalLens-CLIP is in fact a more efficient model design that compares favorably against the baseline FocalLens-MLLM.
>
> **Question 2:** Equipping FocalLens-CLIP with standard MLLM training recipes seems to be an appropriate design. I am curious about the performance. Have you evaluated FocalLens-CLIP's performance on standard multimodal comprehension tasks compared to baseline models? Could you provide results or analysis demonstrating how the context-aware visual features contribute to improved multimodal understanding?
> **Response 2:** In our experiments, we adopt image-text retrieval and linear probing for image classification as the common schemes to evaluate multimodal comprehension. Specifically, we use SugarCrepe and MMVP-VLM datasets for image-text retrieval as they are shown to be much more challenging than standard retrieval benchmarks like MSCOCO. In SugarCrepe, the task is to retrieve the corresponding caption of an image given a correct positive text and a hard-negative caption with only a minor change to the positive caption (e.g., correct caption: “A man with red shirt” → hard negative: “A man with a white shirt ”). While standard CLIP’s features struggle on capturing these subtle image details, we show that we are able to extract finer-grained visual details from FocalLens-CLIP by instructing the model to focus on the “color, patterns and other attributes of the objects” in the image, much improving the performances as shown in Table 6. Similarly, MMVP-VLM dataset tests a model’s capability in understanding subtle image details such as quantity of objects, spatial relationships between them, camera perspective and so on. In Table 7, we show that by specifying FocalLens-CLIP to focus on different aspects of the image of interest (either to focus on quantities, camera perspective of photo, etc.), the resultant context-aware visual features better capture the target image details, leading to significant improvements over standard CLIP features. Finally, in Table 5, we show that context-aware visual features can also make standard linear probing for image classification more data efficient. In particular, in training a dog classifier using ImageNet dog images, we show that the context-aware visual features obtained from FocalLens-CLIP (by specifying “what is the type of the dog”) are able to produce better classifier in a low-data regime compared to standard CLIP features. We conjecture that it is because ImageNet images sometimes contain other objects in the background, and by instructing FocalLens-CLIP to focus on the “dog” in the image, the visual features extracted are less noisy and concentrates more on the dog features.
>
> **Question 3:** Does the training of FocalLens-MLLM still have the next-token-prediction loss based on cross-entropy?
> **Response 3:** We train the baseline FocalLens-MLLM without next-token prediction loss. It would be interesting to have both contrastive loss and next-token prediction loss at the same time. To achieve this in practice, we may need to design more sophisticated mechanisms to handle model outputs differently for computing contrastive loss and next-token prediction loss, as the two objectives may not be directly compatible. We leave further explorations in this direction as future work.

---

> > ### Comment · Reviewer_MEKq · 2024-11-22
> > **Post Rebuttal Comments by Reviewer MEKq**
> >
> > I appreciate the author's rebuttal. After checking the rebuttal, general response, and comments from other reviewers, I have recognized that the scope of this paper remains FocalLens-CLIP. However, my concerns are not addressed. Specifically,
> >
> > 1. Towards Q2, with the community increasingly focusing on MLLMs, the integration of newly designed CLIP models into MLLMs has become a critical performance indicator. For example, DIVA [1] successfully trained a CLIP model that, when integrated into MLLMs, outperformed the original CLIP. Given this precedent, it is highly recommended that we conduct similar experiments to explore the potential improvements in MLLMs.
> >
> > 2. As mentioned in the rebuttal "the entire MLLM is considered a vision encoder", I am curious about the comparison between FocalLens-MLLM and the original LLaVA as vision encoders, since this topic, *i.e.*, regarding MLLMs as visual expert, is quite interesting.
> >
> > **References**
> >
> > [1] Wenxuan Wang, et al. "Diffusion feedback helps clip see better." arXiv preprint arXiv:2407.20171, 2024.

---

> > > ### Author Response · Authors · 2024-11-25
> > > **Further response to Reviewer MEKq (1/2)**
> > >
> > > We thank reviewer MEKq for following up on our reponse and raising further questions. We address the additional questions below.
> > >
> > > **Question 4:** Towards Q2, with the community increasingly focusing on MLLMs, the integration of newly designed CLIP models into MLLMs has become a critical performance indicator. For example, DIVA [1] successfully trained a CLIP model that, when integrated into MLLMs, outperformed the original CLIP. Given this precedent, it is highly recommended that we conduct similar experiments to explore the potential improvements in MLLMs.
> > > **Response 4:** Thank you for bringing this point up. We agree that integrating CLIP models into MLLMs for downstream evaluations is another important performance indicator for CLIP models, as considered in DIVA [1]. Specifically, in DIVA, the authors evaluate CLIP models on two sets of benchmarks: first, on vision-centric image-text retrieval benchmarks such as MMVP-VLM, as in our current evaluations (Table 7); second, on common MLLM benchmarks by training MLLMs with the newly designed CLIP models. Here, we follow the same to first report comparisons of FocalLens-CLIP to DIVA on MMVP-VLM benchmark, and additionally conduct MLLM training experiments to report the results on MLLM benchmarks.
> > > First, on MMVP-VLM, we see that while DIVA on average improves over standard CLIP model (OpenAI ViT-L-14) by an average of 5 points, FocalLens-CLIP further improves over DIVA by another 5 points on average, with significant margins on 5 out of 9 metrics. This shows FocalLens-CLIP is an effective way to improve visual perception capabilities of CLIP models, in addition to relying on external models (e.g., diffusion model in DIVA).
> > >
> > > | Method          | Orientation | Presence | State | Quantity | Spatial | Color | Structure | Text  | Camera | Avg.  |
> > > |------------------|-------------|----------|-------|----------|---------|-------|-----------|-------|--------|-------|
> > > | CLIP            | 6.7         | 20.0     | 26.7  | 6.7      | 13.3    | 33.3  | 46.7      | 20.0  | 13.3   | 20.7  |
> > > | DIVA            | **26.7**        | 20.0     | **33.3**  | 13.3     | 13.3    | 46.7  | 26.7      | 6.7   | 40.0   | 25.2  |
> > > | FocalLens-CLIP  | 6.7         | **33.3**     | **33.3**  | **40.0**     | **26.7**    | **66.7**  |   20.0  |  **26.7** | 20.0   | **30.4**  |
> > >
> > >
> > > Second, we train a LLaVA model variant using FocalLens-CLIP as the vision encoder, and compare its performance to the original LLaVA trained with standard CLIP model. We evaluate the resultant LLaVA models on several MLLM benchmarks, including MM-VET [2], GQA [3], and POPE [4] in the following table. We see that the LLaVA model with FocalLens-CLIP compares favorably to the LLaVA model with standard CLIP, especially on MM-VET that includes tasks requiring fine-grained visual capabilities of MLLMs such as OCR and spatial awareness. These results show another potential application of FocalLens-CLIP as a candidate for training downstream MLLMs for improved visual capabilities.
> > >
> > > | Method                      | MM-VET | GQA   | POPE   |
> > > |-----------------------------|--------|-------|--------|
> > > | LLaVA w/ standard CLIP      | 29.8   | 63.03 | 86.05  |
> > > | LLaVA w/ FocalLens-CLIP     | **31.1**   | **63.56** | **86.10**  |
> > >
> > > [1] Diffusion feedback helps clip see better. Wang et al. 2024.
> > > [2] MM-Vet: Evaluating Large Multimodal Models for Integrated Capabilities. Yu et al. 2023.
> > > [3] GQA: A New Dataset for Real-World Visual Reasoning and Compositional Question Answering. Hudson et al. 2019.
> > > [4] Evaluating Object Hallucination in Large Vision-Language Models. Li et al. 2023.

---

> > > > ### Comment · Reviewer_MEKq · 2024-11-27
> > > > **LLaVA-based Experiments are not sufficient**
> > > >
> > > > I really appreciate the authors' efforts. However, it seems that FocalLens-CLIP brings *marginal* improvements over the standard CLIP.
> > > >
> > > > Moreover, the comparisons are not sufficient. As the authors mentioned that "MM-VET includes tasks requiring fine-grained visual capabilities of MLLMs such as OCR and spatial awareness", could you conduct comparisons on OCR benchmarks (such as OCRBench, ChartQA, TextVQA, and DocVQA) and spatial aware benchmarks (such as MMVP and HallusionBench)? Furthermore, general understanding benchmarks such as MMBench, SEED-Bench, ScienceQA, AI2D, and RealWorldQA are also expected into consideration. Conducting comparisons on such benchmarks is not difficult to implement as open-source tools such as VLMEvalKit and lmms-eval naturally support these benchmarks.

---

> > > ### Author Response · Authors · 2024-11-25
> > > **Further response to Reviewer MEKq (2/2)**
> > >
> > > **Question 5:** As mentioned in the rebuttal "the entire MLLM is considered a vision encoder", I am curious about the comparison between FocalLens-MLLM and the original LLaVA as vision encoders, since this topic, i.e., regarding MLLMs as visual expert, is quite interesting.
> > > **Response 5:** We agree that treating the entire MLLM as a vision encoder that is able to produce image representations is an interesting direction (and that is exactly the motivation for us to explore FocalLens-MLLM). However, since MLLMs like LLaVA are by design for text generations instead of producing explicit representations (embeddings), there is no default way to obtain image representations from LLaVA in a training-free fashion. To consider an alternative way to treat original LLaVA as vision encoders without the introduced contrastive loss in FocalLens-MLLM, we follow recent work [5] to use explicit *prompting* where we instruct LLaVA to generate only a *single token* as its model output, and treat the first output feature of the auto-regressive decoding process as the conditional image feature. For instance, when we are interested in the hair color of the person in the image, we give the following prompt to LLaVA: “What is the hair color of the person? Use one word to answer”. We note this method can be considered as a zero-shot alternative (without contrastive training) to the FocalLens-MLLM baseline. We name this new baseline *LLaVA zero-shot*, and report its performance in the following table, which we will add to our revision.
> > > From the table, we first see LLaVA zero-shot features achieve better performances than unconditional (generic) CLIP features on CelebA-Attribute and GeneCIS datasets, validating that the original LLaVA model does possess strong implicit conditional image features within its LLM decoder, and prompting appears to be an effective approach to extract these features in a zero-shot fashion. However, on fine-grained classification datasets, we see LLaVA zero-shot suffers a significant gap to the standard CLIP features, potentially indicating that the language decoder may be losing some visual information through its forward pass. Additionally, when comparing LLaVA zero-shot to FocalLens-MLLM, we see that while the contrastive loss improves performance on CelebA-Attribute, further training introduces performance drop on other benchmarks, which might be attributed to catastrophic forgetting especially when FocalLens-MLLM is trained only on a smaller scale dataset.
> > > Most importantly, when comparing the new LLaVA zero-shot baseline to our proposed FocalLens-CLIP, we see that FocalLens-CLIP—despite being almost 10x smaller in model size compared to LLaVA—performs competitively across all different evaluations, and significantly better than LLaVA zero-shot on GeneCIS and fine-grained classification tasks. This validates that FocalLens-CLIP is an efficient and promising solution for extracting conditional visual features, and we leave further exploration of extracting visual representations from pretrained MLLMs as a future direction.
> > >
> > > | Method              | CelebA-Attribute | GeneCIS | ImageNet-Subset | Fine-grained Classification Datasets |
> > > |---------------------|------------------|---------|-----------------|--------------------------------------|
> > > | CLIP               | 13.59           | 34.46   | 51.03           | 53.41                                |
> > > | LLaVA zero-shot    | 22.38           | 39.97   | 53.24           | 46.87                                |
> > > | FocalLens-MLLM     | **22.67**           | 37.78   | 52.34           | 32.04                                |
> > > | FocalLens-CLIP     | 21.32           | **43.51**   | **55.29**           | **55.14**                                |
> > >
> > > [5] E5-V: Universal Embeddings with Multimodal Large Language Models. Jiang et al. 2024.

---

### Official Review · Reviewer_Y9rG · 2024-11-03

**Soundness:** 2
**Presentation:** 3
**Contribution:** 2
**Rating:** 5
**Confidence:** 3

**Summary:**

The paper proposes the FocalLens which is a visual feature extraction method that generates different image representations based on the specified context of interest, guided by natural language. By turning a pre-trained vision encoder with vision instruction tuning data, FocalLens emphasizes features relevant to the context, and aims to enhance performance on tasks like image retrieval and classification.

**Strengths:**

The proposed method is validated on multiple downstream tasks for image representations and achieves consistent performance improvements.

**Weaknesses:**

1. The proposed model is unlikely to outperform existing models significantly. The authors mentioned in Lines 139-141 that the proposed model is different from other conditioned vision models (e.g., LLaVA [1], also cited in the paper) because the proposed model can be applied in "broad downstream use cases". However, in the training setting, they use similar training data and settings as LLaVA. This is thus no validation for "being able to do broader downstream tasks".

2. This paper misses the baseline that uses LLaVA features. From the reviewer's understanding, the proposed model looks like a submodule of LLaVA (by removing the language branch). That is, LLaVA is equal to the proposed method if including a text decoder. Currently, the advantage of this work compared with the LLaVA encoding features is unclear.

3. The motivation of this paper is not convincing to the reviewer. In the related works (and some parts of the introduction section), the justification of the difference between existing works and this submission is not clear. The reviewer's understanding is that general MLLM aims to learn general representation, and existing conditional visual representation works aim for task-specific conditioning features. While, this submission aims to learn general conditioning features which might be somewhere between general features and task-specific conditioning features. Then, the question is what is the criterion to distinguish these three features? It is quite confusing what is going to be learned given that related works of conditioning features have been obtained in many existing works. In other words, what is the benefits of learning such in-between conditioning features/representations? It seems general but also specific to some conditions which are not clarified in this paper, given the validate datasets are just common ones.

[1] Visual Instruction Tuning

**Questions:**

1. What is the specific and concrete difference between the proposed method and the existing text-conditioning visual feature exaction method?
2. What kind of broader tasks can be only solved by this proposed method? The contribution (compared with other similar or related works) needs to be highlighted during the rebuttal.

---

> ### Author Response · Authors · 2024-11-20
> **Response to Reviewer Y9rG (1/2)**
>
> **Question 1:** The proposed model is unlikely to outperform existing models significantly. The authors mentioned in Lines 139-141 that the proposed model is different from other conditioned vision models (e.g., LLaVA [1], also cited in the paper) because the proposed model can be applied in "broad downstream use cases". However, in the training setting, they use similar training data and settings as LLaVA. This is thus no validation for "being able to do broader downstream tasks".
> **Response 1:** Thank you for pointing out the confusion. We clarify that by “broader downstream tasks”, we refer to the *types* of downstream tasks (e.g., image classification, image-image retrieval, image-text retrieval) instead of the task domains (e.g., natural images, medical images, artistic images). Specifically, we emphasize that our work aims to build *vision encoders* that are able to generate conditional image representations that can be flexibly used in various downstream applications—such as training classifiers, performing zero-shot classification or retrievals—just like standard CLIP image representations. Although LLaVA’s output text generations are conditioned on both text instructions and images, there is by design no direct way in obtaining text-conditioned image representations from LLaVA. As a result, we consider standard LLaVA as a “text generation” model that implicitly uses conditional visual features, as opposed to the proposed FocalLens models—despite trained on similar data—that produce image representations that can be used in *broader* (as compared to only text generation purpose) range of applications including image-image retrieval, image-text retrieval and image classification, as shown in our experiments. We are eager to provide additional clarification if you have further questions.
>
> **Question 2:** This paper misses the baseline that uses LLaVA features. From the reviewer's understanding, the proposed model looks like a submodule of LLaVA (by removing the language branch). That is, LLaVA is equal to the proposed method if including a text decoder. Currently, the advantage of this work compared with the LLaVA encoding features is unclear.
> **Response 2:** Thank you for the great suggestion! We agree that standard LLaVA features (without the introduced contrastive learning in the baseline FocalLens-MLLM) should also be another baseline to the proposed method FocalLens-CLIP. Since LLaVA is by design for text generations instead of producing explicit representations (embeddings), there is no default way to obtain image representations from LLaVA in a training-free fashion. As a result, we follow recent work [1] to use explicit *prompting* where we instruct LLaVA to generate only a *single token* as its model output, and treat the first output feature of the auto-regressive decoding process as the conditional image feature. For instance, when we are interested in the hair color of the person in the image, we give the following prompt to LLaVA: “What is the hair color of the person? Use one word to answer”. We note this method can be considered as a zero-shot alternative (without contrastive training) to the FocalLens-MLLM baseline. We name this new baseline *LLaVA zero-shot*, and report its performance in the following table, which we will add to our revision.
> From the table, we first see LLaVA zero-shot features achieve better performances than unconditional (generic) CLIP features on CelebA-Attribute and GeneCIS datasets, validating that it is indeed a strong baseline method. The results also show that LLaVA does possess strong implicit conditional image features within its LLM decoder, and prompting appears to be an effective approach to extract these features in a zero-shot fashion. However, on fine-grained classification datasets, we see LLaVA zero-shot suffers a significant gap to the standard CLIP features. On the other hand, we see our proposed FocalLens-CLIP—despite being almost 10x smaller in model size compared to LLaVA—performs competitively across all different evaluations, and significantly better than LLaVA zero-shot on GeneCIS and fine-grained classification tasks. This validates that FocalLens-CLIP is an efficient and promising solution for extracting conditional visual features.
>
> | Method            | CelebA-Attribute | GeneCIS | ImageNet-Subset | Fine-grained Classification Datasets |
> |--------------------|------------------|---------|-----------------|--------------------------------------|
> | CLIP              | 13.59           | 34.46   | 51.03           | 53.41                                |
> | LLaVA zero-shot   | **22.38**           | 39.97   | 53.24           | 46.87                                |
> | FocalLens-CLIP    | 21.32           | **43.51**   | **55.29**           | **55.14**                                |
>
> [1] E5-V: Universal Embeddings with Multimodal Large Language Models. Jiang et al. 2024.

---

> ### Author Response · Authors · 2024-11-20
> **Response to Reviewer Y9rG (2/2)**
>
> **Question 3:** The motivation of this paper is not convincing to the reviewer. The justification of the difference between existing works and this submission is not clear. The reviewer's understanding is that general MLLM aims to learn general representation, and existing conditional visual representation works aim for task-specific conditioning features. While, this submission aims to learn general conditioning features which might be somewhere between general features and task-specific conditioning features. Then, the question is what is the criterion to distinguish these three features?
> **Response 3:** Related to Response 1, we clarify the misunderstanding of the difference between our work and existing related works, specifically (1) unconditional vision encoders (e.g., CLIP) and (2) application-specific models that use implicit conditional visual features (works mentioned in line 130-132). First, we clarify that when compared to existing non-conditional vision encoders (e.g., CLIP models), our goal indeed is to generate representations that are *specific* to the downstream tasks (e.g., understand the camera angle of the photo) than CLIP’s *generic* features. On the other hand, there are existing works that implicitly use conditional visual features for different applications, such as LLaVA and InstructBLIP, which are built for *text generation* purposes. By design, these models do not produce explicit conditional image features, and there is no direct way to obtain this information from models’ hidden representations. Thus, we consider our work as a more *general* approach to produce explicit conditional image representations that can be used in further downstream applications (e.g., classification, retrieval or more). We note the notion of *generality* in this context refers to the compatibility to downstream applications instead of the generality of the data domain that the model is able to tackle (the latter can generally be tackled by scaling up the data for more diverse domain coverage). In short, our work is uniquely positioned against existing works as a “general framework” to extract “task-specific” visual features.
>
> **Question 4:** What is the specific and concrete difference between the proposed method and the existing text-conditioning visual feature extraction method? What kind of broader tasks can be only solved by this proposed method? The contribution (compared with other similar or related works) needs to be highlighted during the rebuttal.
> **Response 4:** Following from Response 3, we do not consider models that implicitly operate on conditional image features (including all works mentioned in related work line 130-141) as visual feature extraction methods as they do not produce explicit image representations compatible for further downstream uses. Instead, the most relevant existing approaches that generate text-conditioning visual features are Composed Image Retrieval (CIR) methods [2,3]. CIR concerns the problem of retrieving target images based on an input query image and a given text condition. As a result, CIR models may generate text-conditioned image embeddings as in FocalLens models. However, our work emphasizes the general notion of “conditional image representations” beyond merely the application of image-to-image retrieval as considered in CIR works. Thus, *for the first time to the best of our knowledge*, we show conditional image representations can enhance downstream performances in multiple settings, ranging from image classification (Table 5), image-text retrieval (Table 6 and 7), to image-image retrieval (Table 2 and 3), where CIR is only one application conditional image representations can be applied to. Importantly, on image-image retrieval tasks which CIR models are designed for, we show FocalLens-CLIP consistently outperforms the state-of-the-art CIR model (i.e., MagicLens) on all considered benchmarks by significant margins.
>
> [2] MagicLens: Self-Supervised Image Retrieval with Open-Ended Instructions. Zhang et al. 2024.
> [3] Pic2Word: Mapping Pictures to Words for Zero-shot Composed Image Retrieval. Saito et al. 2023.

---

> ### Author Response · Authors · 2024-11-25
> **Feedback on our response**
>
> Thank you once again for your valuable review. As the discussion period is close to its end, we wanted to make sure that we have adequately addressed the questions you raised. We would appreciate your feedback on our responses and would love to answer any further questions you have. Thank you!

---

### Official Review · Reviewer_AEGz · 2024-11-03

**Soundness:** 2
**Presentation:** 3
**Contribution:** 2
**Rating:** 5
**Confidence:** 4

**Summary:**

The authors propose a conditional visual feature extraction method that focuses on the representation of specific aspects of the image described in the given text. Specifically, the authors leverage visual instruction tuning data to tune a pre-trained vision encoder in a contrastive manner, taking natural language instructions as additional inputs to produce conditional image representations. Experimental results on a wide range of downstream tasks demonstrate that the proposed method produces features of interest better than generic features produced by standard vision encoders like CLIP.

**Strengths:**

-	The proposed method is well-motivated. The idea of using text instructions as conditions to extract features of interest for certain downstream tasks is intuitive and interesting.
-	The paper is generally well-written and easy to follow.
-	The experiments are extensive, covering a broad range of tasks including image-image retrieval, image classification, and image-text retrieval.

**Weaknesses:**

-	While most results seem promising, some of them are not. For example, in Table 2, FocalLens performs worse than InstructBLIP on GeneCIS (Attribute). In Table 3, FocalLens performs worse than CLIP on Flower and Aircraft. In Table 7, compared with OpenAI ViT-L-14, FocalLens performs the same on Orientation and significantly worse on Structure. These results make me concerned about the actual effectiveness of FocalLens on certain conditions. Could the authors provide a justification on this?
-	The authors use the visual instruction tuning data in LLaVA to train FocalLens models. It would be better to show how the number of visual instruction tuning examples affect the final performance.

**Questions:**

I am concerned about the questions mentioned above. I am leaning towards borderline accept and hope the authors could address my concerns during the rebuttal.

---

> ### Author Response · Authors · 2024-11-20
> **Response to Reviewer AEGz**
>
> **Question 1:** While most results seem promising, some of them are not. For example, in Table 2, FocalLens performs worse than InstructBLIP on GeneCIS (Attribute). In Table 3, FocalLens performs worse than CLIP on Flower and Aircraft. In Table 7, compared with OpenAI ViT-L-14, FocalLens performs the same on Orientation and significantly worse on Structure. These results make me concerned about the actual effectiveness of FocalLens on certain conditions. Could the authors provide a justification on this?
> **Response 1:**  Thank you for pointing this out. We agree that on several occasions, the proposed FocalLens-CLIP is not the best performing method. We provide potential explanations to each of these cases. First, when compared to InstructBLIP, we note that InstructBLIP is trained on much more instruction tuning data than we used to train FocalLens-CLIP. Specifically, in addition to the LLaVA instruction tuning examples, InstructBLIP is additionally trained on a total of 10 other academic datasets. Notably, InstructBLIP is trained on GQA dataset, from which GeneCIS Attribute split is also created from. We conjecture that this may give InstructionBLIP a slight edge in its performance on GeneCIS Attribute due to more similar training and testing data distribution. On the other hand, on GeneCIS Object which is created from COCO images, FocalLens-CLIP performs significantly better than InstructBLIP (both FocalLens-CLIP and InstructBLIP are trained on COCO images). Second, when comparing FocalLens-CLIP to CLIP on classification datasets, while we see slight performance drop on Flower and Aircraft dataset, we also observe improvements on Car and Food datasets. Thus, we consider FocalLens-CLIP to compare favorably to CLIP on standard classification tasks, as also shown by its better average performance than CLIP on ImageNet datasets. Finally, on MMVP-VLM dataset, we observe performance drop on the “Structural Characteristics” tasks. We attribute this to the potential misalignment between the instructions we specified to FocalLens-CLIP and the diverse mix of tasks included in this split. In particular, the Structural Characteristics split includes diverse tasks ranging from “identify the shape of a gingerbread”,  “identify the material of a weight”, to “identify the state of the butterfly wings”. However, regardless of the actual tasks in this split, we only provide FocalLens-CLIP with a generic (and somewhat ambiguous) instruction of “Describe the state of the objects in the image”. As a result, this misalignment between the given instruction and the actual task may lead to observed performance degradation.
>
> **Question 2:** The authors use the visual instruction tuning data in LLaVA to train FocalLens models. It would be better to show how the number of visual instruction tuning examples affect the final performance.
> **Response 2:** Thank you for the great suggestions. We agree that it would be nice to see how the scale of the visual instruction tuning dataset used affect FocalLens-CLIP’s performances. As scaling up the dataset and training the model takes more time, we are still actively working on this and we aim to show preliminary scaling results during the rebuttal.

---

> ### Comment · Reviewer_AEGz · 2024-11-26
> **Official Comment by Reviewer AEGz**
>
> Thanks for the authors' rebuttal. However, the rebuttal does not well address my concerns. The explanations for the inferior performances on certain tasks are not satisfying. When it comes to standard fine-grained classification datasets, the authors only mentioned that "while we see slight performance drop on Flower and Aircraft dataset, we also observe improvements on Car and Food datasets. Thus, we consider FocalLens-CLIP to compare favorably to CLIP on standard classification tasks", but did not provide any insights on why such a phenomenon occurs and how to further improve it. Besides, I have not seen any experiments on how the number of visual instruction tuning examples affects the final performance so far. After reading the comments from the other reviewers, I agree that the current paper does not meet the acceptance threshold and still needs to be further improved. Thus, I am considering lowering my score to 5.

---

> > ### Author Response · Authors · 2024-11-27
> > **Further response to Reviewer AEGz (1/2)**
> >
> > Thank you for reading our response and providing further feedback. We address your additional questions below. In the meanwhile, we believe we have clarified the questions other reviewers raised. We are happy to answer any questions you have regarding to any of those points as well.
> >
> >
> > **Question 3:** When it comes to standard fine-grained classification datasets, the authors only mentioned that "while we see slight performance drop on Flower and Aircraft dataset, we also observe improvements on Car and Food datasets. Thus, we consider FocalLens-CLIP to compare favorably to CLIP on standard classification tasks", but did not provide any insights on why such a phenomenon occurs and how to further improve it.
> > **Response 3:** We apologize for the incomplete answer in our previous response. On fine-grained classification datasets, we conjecture that the performance drop on Flower and Aircraft is mainly due the distribution shift between CLIP’s original pretraining data and the visual instruction tuning dataset (i.e., LLaVA dataset) we used to train FocalLens-CLIP. In particular, it has been observed in the literature that finetuning CLIP on specific data distribution tends to reduce its performances on other datasets [1]. To mitigate this, we conducted an additional experiment where we mix the visual instruction tuning dataset with CC3M [2]—a image-caption dataset commonly used as pretraining dataset for CLIP models—to train our FocalLens-CLIP model. Specifically, we subsample CC3M to a subset of equal size to our visual instruction tuning dataset, where the mixing ratio between the instruction tuning examples and CLIP pretraining-alike examples is 1:1.
> > From the table below, we see that by training FocalLens-CLIP on additional CLIP pretraining-alike examples, we are able to increase FocalLens-CLIP’s performances on both Flower and Aircraft datasets, closing the gap to standard CLIP model. We believe that by more carefully curating the finetuning data distribution or by further scaling, we are able to fully recover CLIP’s performance on these two fine-grained datasets.
> >
> > | Method                                     | Flower | Aircraft |
> > |-------------------------------------------|--------|----------|
> > | CLIP                                       | 83.87  | 25.96    |
> > | FocalLens-CLIP w/ LLaVA dataset           | 80.23  | 21.44    |
> > | FocalLens-CLIP w/ LLaVA dataset + Pretraining data | 82.22  | 22.61    |
> >
> > [1] ​​Robust fine-tuning of zero-shot models. Wortsman et al. 2022.
> > [2] Conceptual Captions: A Cleaned, Hypernymed, Image Alt-text Dataset For Automatic Image Captioning. Sharma et al. 2018.

---

> > ### Author Response · Authors · 2024-11-27
> > **Further response to Reviewer AEGz (2/2)**
> >
> > **Question 4:** I have not seen any experiments on how the number of visual instruction tuning examples affects the final performance so far.
> > **Response 4:** Thank you for bringing this interesting point up and your patience for these additional experiments that take more time. To see how the number of visual instruction tuning examples affects the model performance. we conducted a data scaling experiment where we (a) increase the number of visual instruction tuning examples used to train FocalLens-CLIP, and (b) further mix the visual instruction tuning examples with image-caption pretraining examples as we did in Response 3. In particular, for (a), we increase from 60K examples in LLaVA v1 instruction tuning dataset [3] to LLaVA v1.5 [4] dataset that contains around 600K instruction tuning examples. In the table below, we first see that by scaling up the visual instruction tuning dataset from 60K to 600K, we see significant improvements on CelebA-Attribute and GeneCIS datasets that probe models’ capability to focus on specific image details. However, we also observe that simply scaling instruction tuning examples leads to performance drop on generic classification datasets, including ImageNet-subset and fine-grained classification datasets. This aligns with our previous conjecture that visual instruction tuning examples may introduce distribution shifts that cause performance drop on classification datasets where original CLIP is good at. To remedy this, similar to Response 3, we further mix the 600K visual instruction tuning examples with an additional 600K CLIP pretraining-alike data sampled from CC3M. By training with the additional 600K pretraining data, we see that while there is minor performance drop on CelebA-Attribute and GeneCIS, we are able to much boost the performance on generic classification datasets. From these scaling experiments, we see that the number of visual instruction tuning examples is positively correlated with models’ performance on tasks that require focus on specific image details (CelebA-Attribute and GeneCIS). On the other hand, the more we finetune CLIP model on visual instruction tuning examples, the model may deviate more from its original capability in generic classification tasks. Thus, we demonstrate one mitigation is to mix visual instruction tuning data with CLIP pretraining alike data to strike a balance between the both. We believe that more careful data mixing (with different mixing ratio or more fine-grained distribution selection) may further improve the model’s performance across different tasks.
> >
> > | Method                                  | CelebA-Attribute | GeneCIS | ImageNet-Subset | Fine-grained Classification Datasets |
> > |-----------------------------------------|------------------|---------|-----------------|--------------------------------------|
> > | CLIP                                    | 13.59           | 34.46   | 51.03           | 53.41                                |
> > | FocalLens-CLIP (60K)                    | 21.32           | 43.51   | 55.29           | 55.14                                |
> > | FocalLens-CLIP (600K)                   | 26.34           | 47.73   | 54.98           | 49.41                                |
> > | FocalLens-CLIP (600K + Pretraining data)| 25.48           | 46.51   | 56.59           | 53.53                                |
> >
> > [3] Visual Instruction Tuning. Liu et al. 2023.
> > [4] mproved Baselines with Visual Instruction Tuning. Liu et al. 2023.

---

### Official Review · Reviewer_U695 · 2024-11-04

**Soundness:** 3
**Presentation:** 2
**Contribution:** 2
**Rating:** 5
**Confidence:** 4

**Summary:**

This paper introduces a method called FocalLens, that is designed to improve the visual representation capability of the vision encoders through instruction tuning. The motivation of this method to focus on the specific part of the images, according to the conditions or instructions given. The authors have presented two variants of this method, (i) FocalLens-MLLM : builds upon LLaVA, and (ii) FocalLens-CLIP : builds upon CLIP encoders. The extensive experiments on retrieval tasks have shown superior performance over baselines.

**Strengths:**

The paper focuses on the conditional visual representation of the images, through instruction tuning, which is a good motivation.

**Weaknesses:**

1. The motivation of this paper for retrieval tasks through text instructions, is not a new concept. On the other side, the proposed architectures are similar to LLaVA, with just an addition of contrastive loss, that brings very minor novelty.

2. It is not clear about which instructions are given during the inference of retrieval tasks? It would be better to provide those instructions.

3. I don't see any difference between the textual conditioning of this work and composed image retrieval (CIR) task. As both are shown differently, what are the reasons behind that? A concrete explanation is preferable.

      I think Pic2Word [1] and SEARLE [2], which are focused on CIR tasks, should be one of the proper baselines for the retrieval experiments.

4. LLaVA [3] should also be one of the baseline methods with the LLaVA feature variant of FocalLens.

5. The overview of the apporach is easy to understand, but the overall presentation is not good as the paper lacks the clear explanation of technical details and experimental details.


      [1] Pic2Word: Mapping Pictures to Words for Zero-shot Composed Image Retrieval. Kuniaki Saito, Kihyuk Sohn, Xiang Zhang, Chun-Liang Li, Chen-Yu Lee, Kate Saenko, and Tomas Pfister. CVPR, 2023.

      [2]  Zero-Shot Composed Image Retrieval with Textual Inversion. Alberto Baldrati, Lorenzo Agnolucci, Marco Bertini, and Alberto Del Bimbo. Zero-Shot Composed Image Retrieval with Textual Inversion. In ICCV, 2023.

      [3] Visual instruction tuning. Haotian Liu, Chunyuan Li, Qingyang Wu, and Yong Jae Lee. Advances in neural information processing systems, 36, 2024a.

**Questions:**

See the weakness section. I would like to increase my rating, if the proper justification of my questions will be given.

---

> ### Author Response · Authors · 2024-11-20
> **Response to Reviewer U695 (1/2)**
>
> **Question 1:** Retrieval tasks through text instruction is not a new concept. On the other hand, proposed architecture is similar to LLaVA with contrastive loss.
> **Response 1:** We agree that prior works have explored the problem of image-to-image retrieval with text instructions. Specifically, Composed Image Retrieval (CIR) is perhaps the most related problem setup where the goal is to retrieve target images through composing a given image along with some text instructions. However, in this work, we emphasize the general notion of “conditional image representations” beyond merely the application of image-to-image retrieval as considered in CIR works. In particular, we consider text-conditioned image representations to be like standard image representations that can be used in a variety of downstream applications, including training classifiers, or used directly to perform zero-shot classifications or retrievals. This nuance allows us to demonstrate, *for the first time to the best of our knowledge*, that conditional image representations can enhance downstream performances in multiple settings, ranging from image classification (Table 5), image-text retrieval (Table 6 and 7), to image-image retrieval (Table 2 and 3), where CIR is only one application conditional image representations can be applied to. Under this perspective, our goal is to explore potential ways in generating conditional image representations, where we start from proposing FocalLens-MLLM as a *strong baseline*, and proposing FocalLens-CLIP as our *main method*. We discuss more in detail the comparisons to CIR work in Response 3 below.
>
> **Question 2:** It is not clear about which instructions are given during the inference of retrieval tasks.
> **Response 2:** Thank you for pointing this out. We clarify that all the instructions we used during inference for different tasks are provided in Appendix D (as currently mentioned in Line 253). We will make sure to highlight this for better clarity.
>
> **Question 3:** I don't see the difference between the textual conditioning of this work and composed image retrieval (CIR) task. I think Pic2Word and SEARLE should be one of the proper baselines for the retrieval experiments.
> **Response 3:** Related to Response 1, we first clarify that composed image retrieval (CIR) is one of many applications of conditional image representations considered in our work. In addition, although we consider image-to-image retrieval as one of our evaluations just like CIR, we note that our motivation is different from theirs. Specifically, CIR concerns more on composing the semantics of the input query image with the *external* text condition for retrieving target images with the desired semantics (e.g., retrieving an image of “an origami of goldfish” by inputting an “goldfish” image with the condition “origami”). On the other hand, we focus on extracting visual features that better pronounce certain *observed intrinsic aspects* of the input image using textual instructions as the guidance.
> However, despite the difference in the problem settings and goals, we agree that CIR methods can be considered as  baselines for retrieval experiments. As a result, we did compare our proposed method with MagicLens, which is the current state-of-the-art CIR method shown to outperform Pic2Word and SEARLE on various image-image retrieval tasks [1]. In our experiments (Table 2 and 3), we see FocalLens-CLIP consistently outperform MagicLens on all considered benchmarks by significant margins. We also provide in the Table below comparisons to Pic2Word and SEARLE on GeneCIS dataset, borrowing numbers from [1].
>
> | Method          | GeneCIS-Attribute (R@1) | GeneCIS-Attribute (R@2) | GeneCIS-Attribute (R@3) | GeneCIS-Object (R@1) | GeneCIS-Object (R@2) | GeneCIS-Object (R@3) |
> |------------------|--------------------------|--------------------------|--------------------------|-----------------------|-----------------------|-----------------------|
> | Pic2Word         | 15.7                    | 28.2                    | 38.7                    | 8.4                  | 18.0                 | 25.8                 |
> | SEARLE           | 17.0                    | 29.7                    | 40.7                    | 8.0                  | 16.9                 | 25.6                 |
> | MagicLens        | 16.1                    | 28.2                    | 39.0                    | 16.3                 | 26.2                 | 35.5                 |
> | FocalLens-CLIP   | **19.1**                    | **32.3**                    | **43.3**                    | **20.3**                 | **33.1**                 | **43.7**                 |
>
> [1] MagicLens: Self-Supervised Image Retrieval with Open-Ended Instructions. Zhang et al. 2024.

---

> ### Author Response · Authors · 2024-11-20
> **Response to Reviewer U695 (2/2)**
>
> **Question 4:** LLaVA should also be one of the baseline methods with the LLaVA feature variant of FocalLens.
> **Response 4:** Thank you for the great suggestion! We agree that standard LLaVA features (without the introduced contrastive learning) should also be another baseline in addition to FocalLens-MLLM. Since LLaVA is by design for text generations instead of producing explicit representations (embeddings), there is no default way to obtain image representations from LLaVA in a training-free fashion. As a result, we follow recent work [2] to use explicit *prompting* where we instruct LLaVA to generate only a *single token* as its model output, and treat the first output feature of the auto-regressive decoding process as the conditional image feature. For instance, when we are interested in the hair color of the person in the image, we give the following prompt to LLaVA: “What is the hair color of the person? Use one word to answer”. We note this method can be considered as a zero-shot alternative (without contrastive training) to the FocalLens-MLLM baseline. We name this new baseline *LLaVA zero-shot*, and report its performance in the following table, which we will add to our revision.
> From the table, we first see LLaVA zero-shot features achieve better performances than unconditional (generic) CLIP features on CelebA-Attribute and GeneCIS datasets, validating that it is indeed a strong baseline method. The results also show that LLaVA does possess strong implicit conditional image features within its LLM decoder, and prompting appears to be an effective approach to extract these features in a zero-shot fashion. However, on fine-grained classification datasets, we see LLaVA zero-shot suffers a significant gap to the standard CLIP features. On the other hand, we see our proposed FocalLens-CLIP—despite being almost 10x smaller in model size compared to LLaVA—performs competitively across all different evaluations, and significantly better than LLaVA zero-shot on GeneCIS and fine-grained classification tasks. This validates that FocalLens-CLIP is an efficient and promising solution for extracting conditional visual features.
>
> | Method            | CelebA-Attribute | GeneCIS | ImageNet-Subset | Fine-grained Classification Datasets |
> |--------------------|------------------|---------|-----------------|--------------------------------------|
> | CLIP              | 13.59           | 34.46   | 51.03           | 53.41                                |
> | LLaVA zero-shot   | **22.38**           | 39.97   | 53.24           | 46.87                                |
> | FocalLens-CLIP    | 21.32           | **43.51**   | **55.29**           | **55.14**                                |
>
> [2] E5-V: Universal Embeddings with Multimodal Large Language Models. Jiang et al. 2024.
>
>
> **Question 5:** The overview of the approach is easy to understand, but the overall presentation lacks clear explanation of technical details and experimental details.
> **Response 5:** We apologize that we had to defer some experiment details to appendix due to the page limit. We are however eager to hear your suggestions on adding further clarifications and technical details to the main paper in the revision. Additionally, we aim to publicly release our code for easy reproducibility.

---

> ### Author Response · Authors · 2024-11-25
> **Feedback on our response**
>
> Thank you once again for your valuable review. As the discussion period is close to its end, we wanted to make sure that we have adequately addressed the questions you raised. We would appreciate your feedback on our responses and would love to answer any further questions you have. Thank you!

---

> ### Comment · Reviewer_U695 · 2024-11-27
>
> Thanks to the authors for the rebuttal. It seems from Response 4 that, LLava zero-shot features are much more better than FocalLens-MLLM. On the other hand, although authors highlighted in Response 1, FocalLens-MLLM is a strong baseline, its absence in the table 6 &  7 results both in the main paper and rebuttal is concerning. I am not sure why, but I believe it could be included for comparison.
> Overall, the existence of FocalLens-MLLM still does not bring any novel insights, and FocalLens-CLIP is an innonative approach, which has an interesting motivation and surpasses most of the baselines. The paper is still confusing, and difficult to understand why the proposed models sometimes fail to outperform. I think the paper needs a lot of revision and proper clarifications of the authors' proposal, hence I would like to keep my rating to 5.

---

### Author Response · Authors · 2024-11-20
**General Response**

We thank all reviewers for their valuable comments. They found our study addresses an important problem with clear motivation (Reviewer U695, AEGz, MEKq), and the proposed method is effective and interesting (Reviewer U695, AEGz, Y9rG, MEKq). We clarify and answer all the questions raised by each reviewer below, and will incorporate them into our revision.

**Clarification on the positioning of FocalLens-MLLM:** As there was some common confusion around FocalLens-MLLM, we provide a general clarification on the positioning of FocalLens-MLLM here. First, we would like to reiterate that our goal is to build *conditional vision encoders* able to extract varying visual features from an image based on different user-specified textual-conditions. While existing MLLMs (e.g., LLaVA) can produce different *text generations* for an input image given different instructions, unlike standard vision encoders (e.g., CLIP), MLLMs by design do not produce explicit image representations that can be used for further downstream applications, such as training classifiers or performing retrieval tasks. Thus, to more comprehensively explore potential solutions to our problem of interest, we introduce FocalLens-MLLM as a *stronger baseline approach* than unconditional CLIP models, for the purpose of enriching the completeness of this study. On the other hand, we consider FocalLens-CLIP to be the *main proposed method* in this work given its relative efficiency and performance advantages compared to FocalLens-MLLM as mentioned in Section 4.3. We apologize for the confusion and will make sure to clarify the baseline and the main proposed method in our revision.

---

### Meta-Review · Area_Chair_9cfd · 2024-12-20

**Metareview:**

While many existing pre-trained models, e.g., CLIP, can extract visual features from images, this work investigates conditional feature extraction that can obtain different visual representations according to text instructions. Many reviewers find the direction interesting, but they have more concerns about novelty, effectiveness and model design. Therefore, all reviewers rated the work as 5 after discussion. AC encourages authors to improve the work with comments from reviewers.

**Additional Comments On Reviewer Discussion:**

After reading the rebuttal, Reviewer U695 found that FocalLens-MLLM lacked novelty and the work was still confusing. Reviewer AEGz decreased the score to 5 since the concerns were not fully addressed. Reviewer MEKq was also unsatisfied with the rebuttal and concerns remained. All reviewers agree that the work in the current form is below the acceptance threshold.

---

### Decision · Program_Chairs · 2025-01-22

Reject